# Does Mexico Have Enough Land to Fulfill Future Needs for the Consumption of Animal Products?

**Maria-Jose Ibarrola-Rivas [1],***  **and Sanderine Nonhebel [2]**

[1]  Departamento de Geografía Física, Instituto de Geografía, UNAM, Mexico City 04510, Mexico
[2]  Energy and Sustainability Research Institute Groningen (ESRIG), University of Groningen, 9747AG Groningen, The Netherlands; s.nonhebel@rug.nl
*  Correspondence: ibarrola@igg.unam.mx, Tel.: +52-55-5622-4240 (ext. 45507)

**Abstract:** Land demand arising from the consumption of animal products is one of the greatest challenges for future sustainability. Developing countries are changing rapidly in both the consumption of animal products and the livestock production systems. Mexico is used as an example of a developing country. An approach is developed to identify the production variables that drive the Land Requirement for Animal Products (LRAP) for beef, milk, pork, chicken meat, and eggs. An average medium-scale farm of Mexico is described using farm-scale production data from the National Agricultural Survey of Mexico. The results show that the use of grassland outweighs the use of cropland for feed production, and the use of barn area is least. The production of beef protein requires more land than any other animal product because of its large demand for pasture land. The use of grassland represents 70% of the total demand for land for food by the Mexican population, and this is mainly for beef and milk consumption. Population growth and changes to a more affluent diet will result in a demand for more land for food; however, there will not be enough land if food is produced with present livestock production systems. It is necessary to implement strategies to reduce the use of land for food by focusing on both production and consumption.

**Keywords:** land use; Mexico; animal products; management livestock systems; impact of food consumption

## 1. Introduction

The consumption of animal products is the main challenge for sustainability of the global food system [1–3]. This problem will increase owing to the on-going dietary changes towards more affluent consumption mainly in transitional and developing countries [4,5]. Land demand for future generations can only be met by both improving production systems and reducing globally the consumption of animal products in the diet [2,3], or by shifting the type of protein intake, e.g., from beef to pork or poultry, or from beef to legumes [3]. The high demand for land by animal production is due to the inefficiency of production of one animal protein unit compared with production of one vegetable protein unit: three to 10 times more land is needed to produce a meat protein than a vegetable protein [6].

Agricultural statistical databases [7] supply data on nationally available agricultural land, cropland and grasslands, and also national domestic food and feed consumption. However, some of the food and feed produced domestically is exported, and some of the food and feed consumed domestically is imported. Additionally, agricultural statistics do not specify which agricultural area is used for each crop or animal product. Therefore, a detail analysis of the use of land for each food item is required.

Calculation of the requirements of land is less straightforward for animal products than for crop-based food. For the latter, the land requirements can be calculated with the crop yield: kilograms of crop produced per hectare. In contrast, the land needed for animal products is the land used by the

animal during its lifetime, which is mainly the land for its feed [8]. This land is difficult to quantify because the type and amount of feed consumed depends on production variables such as the type of animal, the age of the animal, and the price of the feed [8]. Additionally, some animals (cattle) require grasslands for which production data are not available; for instance, the number of animals per grassland area is not a common variable in the statistical databases.

　　Some studies have quantified the land use for animal products [8–14]. Land for animal products is the land used for feed production (cropland for feed and grassland). In general, the cropland for the feed that livestock consume is calculated assuming a feed conversion factor [8], but for grassland, it is more complex. Some studies have calculated the grassland area for animal products based on available statistics [9,10] or on estimations or modeling of grassland productivity [11,12]. No empirical data exist on the use of grassland by each type of livestock [9]. Nijdam et al. [13] presented an extensive review of studies that calculated the land required to produce one kilogram of animal protein; these studies show that large differences exist in the land requirements not only among the different types of animal products, but also among different types of production systems (ranging from extensive to intensive).

　　In Mexico, consumption and production of animal products has been rapidly changing in recent decades. The average per capita consumption of animal products doubled in the period 1961-2013, thereby increasing more rapidly than the global average which increased by "only" 50% [7]. The per capita consumption of beef and milk increased by 80%, and that of chicken meat and eggs increased nine fold and six fold, respectively. At the same time, the livestock industry markedly increased in response to not only the increase in per capita consumption, but also to the increase of the population and the increase of exports. During this period, total meat production increased from 1 to 6 million tons per year [7], and exports of meat increased from 30 thousand to 300 thousand tons per year [7]. It is expected that this trend will continue, since population and consumption will increase in the coming decades. Most studies of land use for livestock products have been in developed countries [8–14]. Mexico is a good example to illustrate the livestock production situation in a developing country that is facing economic transition. It is necessary to understand the impacts of these livestock systems to identify pathways to reduce environmental impact in the future. These insights can be useful for other countries that are facing rapid changes in both diets and livestock production systems.

　　The aim of this paper is to determine how much land is needed to produce the animal products consumed by the Mexican population and to estimate how this value will change in future. To do this, we analyzed the livestock production variables that drive the land requirements to produce animal products in Mexico, we discuss implications for future food supply, and we identify options to reduce the land requirements for the demand for food in Mexico. The study considered the five main animal products for Mexican consumption, which account for 84% of the total average per capita protein intake: beef (15%), milk (24%), pork (11%), chicken meat (21%), and eggs (24%) [7]. We calculated the land required to produce these five animal products in a typical medium-scale farm with farm-scale production data. These values were scaled up to discuss the demand for land for food in Mexico in terms of diverse dietary patterns in 2050. Finally, options to reduce land use were discussed on the basis of (1) the role of the main livestock production variables and (2) the role of diets.

## 2. Materials and Methods

### 2.1. Theoretical Framework: Land Use Associated with Livestock Systems

　　The Land Requirement for Animal Products (LRAP) is the land needed, in m$^2$, to produce 1 kg of meat, milk, or eggs (Figure 1). For meat, this is the land that the livestock animals use during their lifetime; for milk and eggs, it is the land that animals use to yield a certain amount of food. The amount of land depends on different variables intrinsic to the type of animal and to the type of production system.

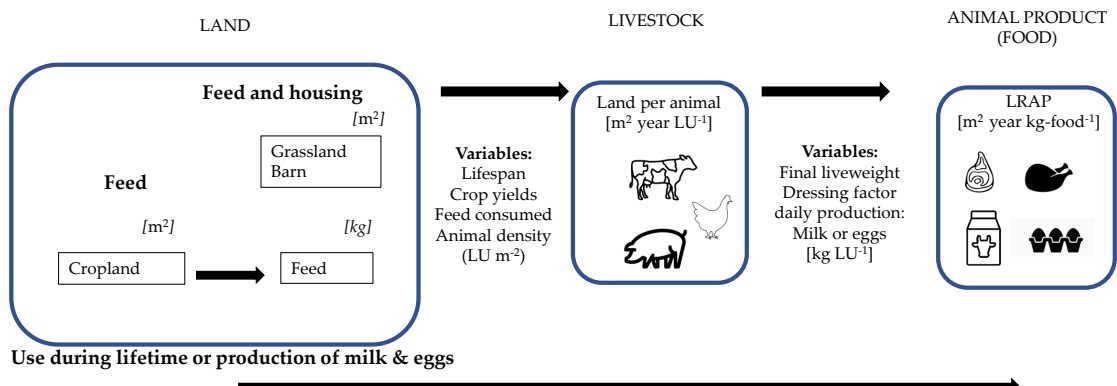

**Figure 1.** Calculation of the Land Requirements for Animal Products (LRAP). Source of icons: the Noun Project [15].

Animals mainly use land for housing (barns) and for their feed. The land for feed is the cropland used to produce the feed that animals consume. Some intensive systems use only indoor housing for cattle and do not use grassland directly; they are fed on grains and grass, and their LRAP is mainly indirect cropland to produce the feed that the animal consumes during its lifetime [8]. In contrast, some extensive systems raise ruminants mainly on grasslands, thus, they directly use the land for their grass consumption [16].

The land for housing depends on the number of animals per area of farm: animal density. This density is calculated in terms of the number of Livestock Units (LU) on the farm. The number of Livestock Units is adjusted by the age of the animals; for instance, a young calf counts as one-half of a Livestock Unit. The Livestock Unit density varies among types of systems (intensive barn systems versus extensive systems) and among types of animal (e.g., a greater number of chickens than of cows can be accommodated in a barn).

The land for feed can differ for several reasons:

(1) Each animal requires different amounts and types of feed (due to nutritional requirements) depending on the livestock species, breed, age, physiological state, physical activity, etc. [8].
(2) Intensive indoor systems and extensive grazing systems require different amounts and types of feed.
(3) Crop yields of the feed (ton ha$^{-1}$) differ depending on where the feed was produced [8] and from year to year [7].
(4) Feed available on the market can vary monthly in price, in nutritional value, and in the mix of imported crops and domestically produced crops [7,8].

Calculation of the LRAP needs to consider the duration of land use by the animal. For instance, if a farmer has a certain barn area, the farm can produce more meat per year with a species that uses the land for only 2 months than with a species that uses the land for 2 years. In other words, if animals of different species use the same amount of land to produce 1 kg of meat, the land requirement should have a different value if the land was used for 2 years than if it was used for 2 months. For this reason, the LRAP is expressed in "annual square meters", which is the area used during a year. Cows, chickens, and pigs have different lifespans to reach an optimal commercial weight. Longer lifespans result in greater land use because (1) animals use land longer (barn or grassland) and (2) Animals need more feed to maintain metabolic functions and to put on weight so the conversion of feed to meat obtained is inefficient.

The type and amount of feed consumed by each animal differs owing to different nutritional requirements, and animals gain weight differently per kilogram of feed consumed, thereby reaching an optimal weight after different periods. In relation to the land for feed, the cropland is used during the whole growing season of the crop: from planting to harvest. Sometimes, two harvests are possible

during the year, which doubles the duration of use of the cropland area. In this paper, we used data for annual crop yields, the production per area per year.

*2.2. Data Source*

Farm-scale production data are needed to calculate the LRAP (see variables in Figure 1). The main data source for this paper was farm-scale livestock production data from the National Agricultural Survey of Mexico for 2014 [17], performed by the National Institute of Statistics and Geography of Mexico (INEGI). We used the microdata of this survey which represent the responses of the farmers, and therefore, these are farm-scale data. We aggregated the farm-scale data to a national scale to discuss the national relevance. The microdata were accessed and processed at the Microdata Laboratory of INEGI [18] with project number LM-530.

When production data were not available, this survey used handbooks for livestock farmers; they give proxies for average Mexican livestock production systems [16,19–23]. For beef and milk production systems "The handbook of cattle for meat" [16] and the "The handbook of cattle for milk" [19] were used for the data of barn area per animal, the feed composition and the animal's food intake daily requirement. For pork production systems, the publication from the Agricultural Ministry of Mexico (SAGARPA) called "Swine Production System, Technical information sheet 11" [20] was used for the data of feed composition, and the "Swine handbook" [21] was used for the barn area per pig depending on the age of the animal. For chicken meat and egg production, the "Poultry handbook" [22] was used for the barn area per chicken and feed composition, and the recommendation handbook "With homemade concentrates improve the feeding of your birds and increase production" [23] was used for the average final weight of bird for meat production and the requirements of feed per animal.

Crop yield data of the main feed crops were used to calculate the indirect land for feed. The data sources were both the Statistics of the Food and Agricultural Organization of the United Nations (FAO) [7] and the Mexican National Information Service of Agrofood and Fisheries of the government (SIAP) [24].

The agricultural survey [17] gathered detailed livestock production values from 23,118 cattle farmers, 7849 pig farmers and 237 poultry farmers in 2014. These producers were a representative sample of the total livestock producers of the country.

In Mexico, production systems range from small-scale livestock systems with only few animals per farmer to large-scale systems with a large number of animals per farmer. Each system differs in terms of production strategies and productivity in terms of meat, egg, and milk production per animal. For this study, we use an average medium-scale farm to represent a typical Mexican farm and used its production variables to calculate the LRAP. To identify this "average medium-scale farm", we grouped the livestock farmers reported by INEGI [17] on the basis of number of animals per farmer. We selected the following farmers to characterize the "medium-scale farm": for milk and beef production, farmers with 20 to 50 animals; for pork production, farmers with 10 to 50 pigs; and for chicken and egg production, farmers with 1000 to 10,000 chickens. The production data of the farmers of each system were aggregated to yield average production variables for each livestock system, and these were used to calculate the LRAP. The numbers of "medium-scale farms" from which data were used for the calculation were as follows: 2025 farms for beef production, 978 farms for milk production in pasture systems, 214 farms for milk production in barn systems, 1240 farms for pork production, six farms for chicken meat production and five farms for egg production. For chicken meat and eggs, the "poultry survey" associated with the Agricultural Survey [17] was used; this has detailed production data for eggs and chicken meat, but a smaller sample of farms.

## 2.3. Calculations of LRAP

### 2.3.1. Land for Feed

The four main feed crops in Mexico in 2014 were maize and products (58%), sorghum and sorghum products (34%), soybeans (4%), and barley [7]. These feed crops, in addition to sorghum forage, which is widely used as bovine forage according to SIAP [24], were used to calculate the land for feed (Table 1). Each animal needs a different mix of feed, which is described in Section 2.4. The land for feed depends on the annual crop yield of each crop. A large share of the national feed supply is imported, thus, we calculated a weighted crop yield for each feed crop considering the share of imports in 2014 [7]. We assumed that imported feed comes from the USA since it is the main source of feed imports to Mexico [25]. No data were available for imports of sorghum forage, thus, we assumed all sorghum forage is produced domestically; to derive the dry matter value of the forage crop, we assumed this to be 47% of the green sorghum forage [26]. The Mexican green sorghum forage crop yield in 2014 was 29 ton ha$^{-1}$ [24].

**Table 1.** Average crop yield for the main five feed crops and the contribution of imports. Source data: Food Balance Sheets (imports) [7]; crop yield values of 2014 from FAO [7] and SIAP [24].

|  | Maize Grain | Sorghum Grain | Barley Grain | Sorghum Forage | Soybeans |
|---|---|---|---|---|---|
| Share of imports (in respect to national supply) | 23% | 16% | 49% | – | 94% |
| Mexican crop yield [ton ha$^{-1}$ year$^{-1}$] | 3.3 | 4.2 | 2.7 | 13.6 (dry matter) | 1.8 |
| USA crop yield [ton ha$^{-1}$ year$^{-1}$] | 10.7 | 4.2 | 3.9 | – | 3.2 |
| Weighted crop yield considering imports [ton ha$^{-1}$ year$^{-1}$] | 5.0 | 4.2 | 3.3 | 13.6 | 3.1 |
| Land for Feed [m$^2$ year kg-feed$^{-1}$] | 2.0 | 2.4 | 3.0 | 0.7 | 3.2 |

### 2.3.2. Dressing Factor of Livestock Animals

In the case of meat production, the final live weight of the animal is not the total amount of meat because the live weight includes fluids, bones, and other non-edible materials. The carcass weight is the dressed body of a meat animal. The ratio between the carcass weight and the live weight is the so-called "dressing factor", which determines the final amount of meat in relation with the animal's weight when slaughtered. The dressing factor differs between cattle, pigs, and chickens (Table 2). We calculated the dressing factors for all meat products using the average carcass yield reported by the FAO [7] and the live weight reported by INEGI [17] (Table 2).

**Table 2.** Dressing factors for beef, pork and chicken meat calculated for this paper. Source of data: live weight of the Livestock Unit (LU) [17]; carcass weight per Livestock Unit: national average in 2014 from the Food Balance Sheets [7].

|  | Live Weight when Slaughtered [kg LU$^{-1}$] [17] | Carcass Weight [kg LU$^{-1}$] (Yield Carcass Weight from [7]) | Dressing Factor [Carcass-Weight × Live-Weight$^{-1}$] |
|---|---|---|---|
| Beef | 539.3 | 212.3 | 0.39 |
| Pork | 87.5 | 78.5 | 0.90 |
| Chicken meat | 2.3 | 1.78 | 0.77 |

### 2.3.3. LRAP

Equations (1)–(6) were used to calculate the LRAPs, which include the production variables shown in Figure 1. The units of the elements of the equations are indicated below each equation. The calculation of LRAP is different for meat, milk, and eggs. For meat, we calculated all the land used by the animal during its lifetime. For the production of milk or eggs, we calculated the land needed by the animal during the period associated with that production; we calculated the LRAP in relation to the annual production of milk or eggs per animal. We did not include the periods of non-production, such as when the animals are too young, or when the cow is calving.

**MEAT:**

$$Housing\ Land_{meat} = (Barn\ area \times lifespan)$$
$$\left[m^2 \cdot year \cdot LU^{-1}\right] = \left(\left[m^2 \cdot LU^{-1}\right] \cdot [year]\right) \tag{1}$$

$$Feed\ Land_{meat} = Land\ for\ feed \times Feed\ consumed\ + Grassland \times period\ used$$
$$\left[m^2 \cdot year \cdot LU^{-1}\right] = \left[m^2 \cdot year \cdot kg\ feed^{-1}\right] \cdot \left[kg\ feed \cdot LU^{-1}\right] + \left[m^2 \cdot LU^{-1}\right] \cdot [year] \tag{2}$$

$$LRAP_{meat} = Live\ weight \times dressing\ factor^{-1} \times \left(Housing_{meat} + Feed_{meat}\right)$$
$$\left[m^2 \cdot year \cdot kg\ meat^{-1}\right] = [kg\ live\ weight] \cdot [kg\ meat \cdot kg\ live\ weight^{-1}]^{-1} \cdot \left(m^2 \cdot year \cdot LU^{-1} + m^2 \cdot year \cdot LU^{-1}\right) \tag{3}$$

**MILK AND EGGS**

$$Housing\ Land_{milk\ and\ eggs} = (Barn\ area)$$
$$\left[m^2 \cdot LU^{-1}\right] = m^2 \cdot LU^{-1} \tag{4}$$

$$Feed\ Land_{milk\ and\ eggs} = (Land\ for\ feed \times Annual\ feed\ consumtion) + Grassland$$
$$\left[m^2 \cdot LU^{-1}\right] = \left[m^2 \cdot year \cdot kg\ feed^{-1}\right] \cdot \left[kg\ feed \cdot LU^{-1} \cdot year^{-1}\right] + \left[m^2 \cdot year \cdot LU^{-1}\right] \tag{5}$$

$$LRAP_{milk\ and\ eggs} = annual\ production\ per\ animal^{-1} \times \left(Housing_{milk\ and\ eggs} + Feed_{milk\ and\ eggs}\right)$$
$$\left[m^2 \cdot year \cdot kg\ food^{-1}\right] = \left[kg\ food \cdot LU^{-1} year^{-1}\right]^{-1} \cdot \left(\left[m^2 \cdot LU^{-1}\right] + [m^2 \cdot LU^{-1}]\right) \tag{6}$$

Note: the second part of Equation (2) applies only to the beef production system. Beef cattle go through two management periods during their lifetime: grassland and fattening period (see Section 2.3). For both periods, animals consume feed but in different amounts. This equation is used for both management periods, and the Land for feed which is used in Equation (3) is the sum of the two periods.

### 2.4. Livestock Systems

### 2.4.1. Beef Production

The systems for beef production generally consist of a grazing period followed by a fattening period during which the animals are inside a barn (Figure 2). The purpose of the grazing period is that the animals grow slowly and are cheap in terms of production costs. To calculate the LRAP, we calculated separately with different samples of farms for the grazing period and the fattening period (for both Equations (1) and (2)). For the grazing period, we used the middle-scale farms that only use outdoor grazing, which accounted for 1945 farms [17]. For the fattening period, we used the middle-scale farms that only raise the stock in barns, which accounted for 80 farms [17]. The source agricultural survey does not specify the duration of the grazing period, therefore we assumed it to be 1.5 years. The fattening period consists of 5.2 months, to reach 539 kg per animal [17]. The final amount of beef produced by each animal is the final live weight multiplied by the dressing factor.

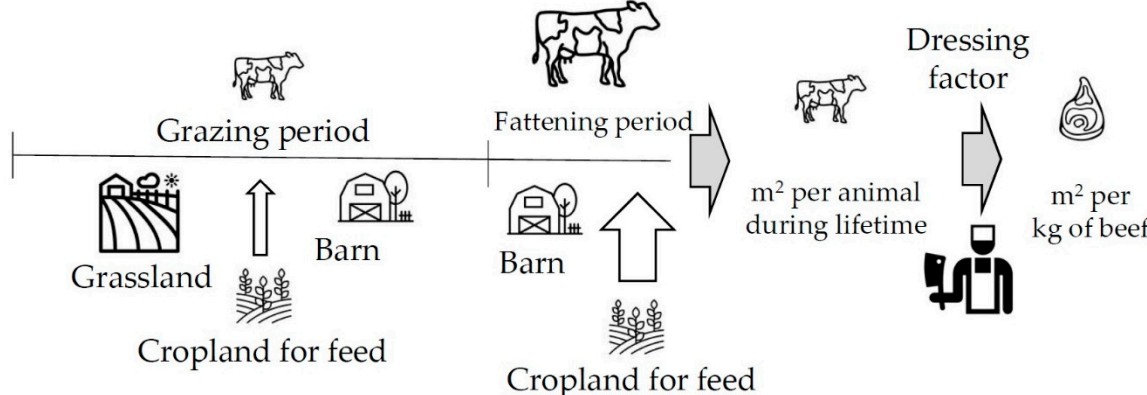

**Figure 2.** Beef production system. Source of icons: the Noun Project [15].

To calculate the barn area, we assumed that each animal uses 12 m$^2$ of barn during its lifetime (during the grazing and fattening period) [16] (p. 77).

The grassland area used per LU depends on the duration of the grazing period and on the carrying capacity of the grassland. The carrying capacity of the grassland is the number of LUs that a certain grassland can support, depending on conditions such as climate, soil chemistry, etc. The number of LUs achieved per area of grassland is not always the same as the theoretical carrying capacity of the grassland, because of social, economic, and technological limitations of the farmers. In the national survey [17], each farmer reports the area of grassland that they use (whether it is enclosed area, natural grasslands, or planted grasslands), and the number of animals and their age that they have in that area (Table 2). Farmers have cattle of diverse ages: calves, bullocks to grow for beef production, bulls for fertilization, etc. Carrying capacity varies with age: young animals count less than older ones. We calculated the LUs of each farm based on the age of each animal, assuming that an animal younger than 1 year old is equivalent to 0.4 LU, one between 1 and 2 years old is 0.7 LU, and one older than 2 years is 1 LU [27]. Based on these values, the head of cattle per area of grassland for the medium-scale farms analyzed in this paper is 0.20 LU ha$^{-1}$.

The amount of feed that the animal consumes depends on its live weight, and this differs between the grazing and fattening periods. During the grazing period, bullocks consume daily 2% of their live weight of feed, and during the fattening period, they consume 3% of their live weight [16]. During the grazing period, farmers report that only 32% of the bullocks are fed with grains and 55% are fed with forage crops [17], thus, we used these values to calculate the amount of feed consumed during the grazing period. Farmers do not report the weight of the animals when they start the fattening period, thus, we assumed that the animals reach half of their final weight after the grazing period, and that during the grazing and fattening period their growth rate is linear. With these assumptions, we calculated an average monthly live weight that we used to calculate the amount of feed consumed per month during the grazing and fattening periods. We assumed that during the fattening period, all animals consume the recommended amount, and that during the pasture period, they consume 32% and 55% of the recommended amounts of grains and of the forage crops, respectively, on the basis of data reported by INEGI [17]. Finally, we calculated the amount of feed consumed during the pasture and the fattening period, which is used in Equation (2).

Feed is usually a mix of grains and forage. According to Lesur [16], feed should consist of 30% forage and 70% grains, and the grains should be 60% maize grain, 30% sorghum grain, and 10% barley grain. The forage is usually sorghum forage. These percentages are with respect to "weight" of the total feed consumed by the animal. Equation (7) was used to calculate the Land for Feed for Equation (2) using the "weighted" crop yields of Table 1.

$$Land \ for \ Feed = \sum_n A \times B \tag{7}$$

where *B* is the crop yield of the feed crop or forage crop, and *A* is the share of that crop in the mix of the feed. *n* is the number of feed crops that contain the mix. The units of the Land for Feed are $\left[m^2 \cdot year \cdot kg\ feed^{-1}\right]$.

### 2.4.2. Milk Production Systems

Cows for milk production can be maintained indoors or on grasslands (Figure 3). The farms that use grazing systems also give grain and forage feed to the cows. According to the farmers [17], 59% of the cows are given additional grain feed and 67% are given additional forage feed. The number of cows per hectare of pasture in the milking systems is 0.47 LU ha$^{-1}$. This value is higher than the livestock density of the grazing period of the beef production system, but the additional feed allowance is larger in the milking systems.

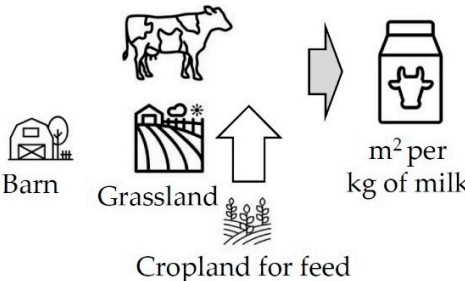

**Figure 3.** Milk production systems. Source of icons: the Noun Project [15].

The average milk production is 14 kg LU$^{-1}$ day$^{-1}$ in the barn system and 6.7 kg LU$^{-1}$ day$^{-1}$ in the grazing systems [17].

We used Equations (4)–(7) to calculate the LRAP for milk production. We assumed that the daily production is evenly distributed throughout the year. As with the beef production systems, we assumed that an LU uses 12 m$^2$ of barn during the lifetime in both the barn and the grazing systems [16].

The consumption of feed per cow was calculated as follows. We assumed that a milking cow weighs 600 kg. As with the beef production systems, we assumed that a grazing cow consumes 2% of its live weight in feed and a cow in a barn system consumes 3% of its live weight in feed [19].

### 2.4.3. Pork Production

In Mexico, pigs are housed in barns (Figure 4). The "Feed Land" for pork is only the cropland for feed production. Farmers report the average pig's age before slaughter as 199 days (herein referred to as "lifespan"), and their live weight is 88 kg [17].

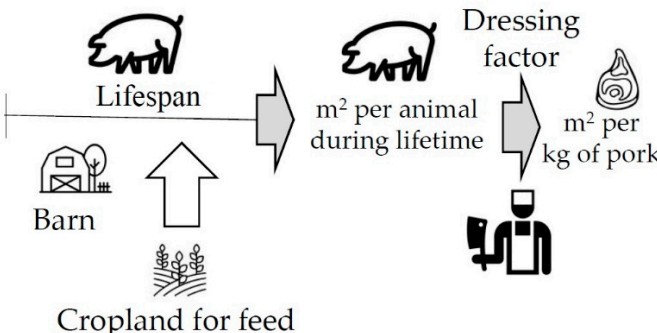

**Figure 4.** Pork production systems. Source of icons: the Noun Project [15].

The daily amount of commercial feed given to the pigs is 2.2 kg $LU^{-1}$ $day^{-1}$ [17]. However, not all farmers give commercial feed; according to INEGI [17], although 84% of pigs are fed with commercial feed, the others might be fed with food waste or agricultural waste. We considered this percentage in our calculations by multiplying 84% by the total feed consumed per LU (2.2 kg $LU^{-1}$ $day^{-1}$) and by the lifetime (199 days).

According to the SAGARPA information sheet [20], pig feed consists of 80% sorghum grain, 10% soya paste, and the rest is calcium, salt, and other minerals. For our calculations, we assumed 85% of the feed to be sorghum grain and 15% to be soybeans, and we considered the share of imported grains (see Table 1).

The barn area needed per pig depends on the weight (or age) of the pig (Table 3). Pigs require less barn area when they are younger, and their requirement of space increases as their weight increases [21]. We used the values of Table 3 and Equation (8) to calculate the average barn area used per pig during its life.

**Table 3.** Barn area required per pig based on live weight and age. Data source: [21].

| Live Weight of the Pig | Age [month] | Barn Area [$m^2$ $pig^{-1}$] |
|---|---|---|
| Mother sow with piglets | 1 | 0.1825 |
| 4.5 kg | 2 | 0.185 |
| 13.6 kg | 3 | 0.3 |
| 33 kg | 4 | 0.46 |
| 67.5 kg | 5 | 0.65 |
| 90 kg | 6 | 1.4 |

Finally, the dressing factor is 0.90 (Table 2).

$$\text{barn area per LU}\left[m^2 barn \times LU^{-1}\right] = \left(\sum_{n=1}^{n=6} barn\ area_n\right) \times 6^{-1} \tag{8}$$

### 2.4.4. Chicken Meat and Egg Production

In Mexico, chickens for meat and egg production are reared indoors, thus, the land for feed is only the cropland for the feed production (Figure 5). For egg production, two types of production practices were analyzed: "on foot farms", where chickens are free on the ground of the barn; and "cage farms", where chickens are in cages [23].

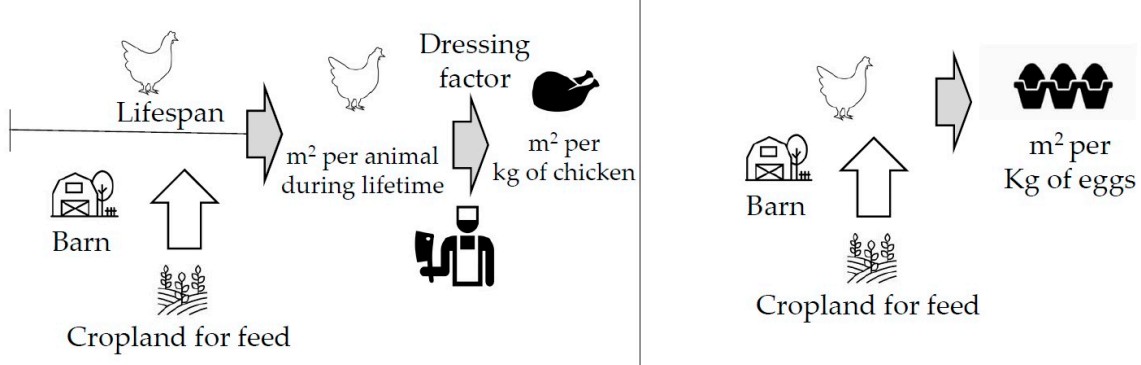

**Figure 5.** Chicken meat and egg production systems. Source of icons: the Noun Project (2018).

We used Equations (1)–(3) to calculate the LRAP for chicken meat and Equations (4)–(6) to calculate LRAP for eggs. For meat production, the average barn area needed is 0.09 $m^2$ $LU^{-1}$ [22]. For egg

production (laying hens), the barn area per hen is 1.39 m$^2$ LU$^{-1}$ for "on foot" farms and 0.47 m$^2$ LU$^{-1}$ for caged chickens [22].

According to Lesur [22], feed for chickens should be a mix of maize (66%), soya (13%), and minerals and other grains. We assumed the mix for this study to include only maize (75%) and soybeans (25%). The feed requirement per chicken is 1.14 kg week$^{-1}$ [23]. We assumed chickens in all production systems consume the same amount and type of feed. We used the weighted crop yield considering the share of imports (Table 1). The productivity of the chicken meat systems depends on the number of fattening cycles that farmers use in one year; they report 4.2 fattening cycles per year [17], which means that the chickens live, on average, 2.8 months. Egg production farmers report an average daily production of 0.07 kg eggs LU$^{-1}$ day$^{-1}$.

Finally, we assumed that all chickens for meat production weigh 2.3 kg when slaughtered [23] with a dressing factor of 0.77 (see Table 2).

## 2.5. Production Variables and Nutritional Relevance

Table 4 shows the summary of production variables for the five animal products that are needed to calculate the LRAP.

**Table 4.** Variables that determine the LRAP for the livestock production systems. Data sources: [1] [16]; [2] calculations with Equation (8) using [21]; [3] [22]; [4] [20]; [5] calculations by the authors using [19]; [6] calculations by the authors using [17]; [7] [23]; [8] calculations by the authors using FAO statistics [7]; [9] calculations by the authors; [10] assumption for this paper.

| Variable | Beef | Milk Grazing | Milk In Barn | Pork | Chicken Meat | Eggs On Foot | Eggs Caged |
|---|---|---|---|---|---|---|---|
| **Housing** | | | | | | | |
| Barn area per Livestock Unit [m$^2$ LU$^{-1}$] | 12 [1] | 12 [1] | 12 [1] | 1.06 [2] | 0.09 [3] | 1.39 [3] | 0.47 [3] |
| **Feed** | | | | | | | |
| Type of feed | Maize grain, sorghum (grain and forage), and barley grain [a,(1)] | | | Sorghum grain and soybeans [4] | Maize grain and soybeans [3] | | |
| Daily amount of feed [kg LU$^{-1}$ day$^{-1}$] | 1 (grazing period) and 12 (fattening period) [b,(5)] | 7.4 [5] | 18 [5] | 2.2 [6] | 0.16 [7] | | |
| Total amount of feed during lifetime [kg LU$^{-1}$] | 2 836 [5,6] | — | — | 368 [6] | 14 [6,7] | — | — |
| Crop yield [ton ha$^{-1}$] [c] | 7.3 [8] | | | 4.0 [8] | 4.5 [8] | | |
| Grassland carrying capacity [LU ha$^{-1}$] | 0.20 (grazing period) [6] | 0.47 [6] | — | — | — | — | — |
| **Productivity variables** | | | | | | | |
| Lifespan [yrs] | 1.9: 1.5 [10] (grazing period) + 0.4 [6] (fattening period) | — | — | 0.54 [6] | 0.24 [6] | — | — |
| Live weight when slaughtered [kg LU$^{-1}$] | 539 [6] | — | — | 88 [6] | 2.3 [7] | — | — |
| Dressing factor [e] | 0.39 [9] | — | — | 0.9 [9] | 0.77 [9] | — | — |
| Daily productivity [kg-food LU$^{-1}$ day$^{-1}$] | — | 6.7 [6] | 14 [6] | — | — | 0.07 [d,(6)] | 0.07 [d,(6)] |

[a] The mix varies between grazing and fattening periods. See text for details. [b] Daily consumption by beef animals depends on age, live weight and whether it is the grazing period or fattening period. These values are the average during the whole grazing or fattening period. Thus, the total consumption of feed during the period is divided by the duration of the period. See details in the description of the beef production systems. [c] These crop yields are the weighted calculated crop yield of the mix: for bovines, 30% forage and 70% grains (maize, sorghum and barley); for pork, 85% sorghum grain and 15% soybeans; for chicken, 75% maize and 25% soybeans. [d] Farm-scale data from National Institute of Statistics and Geography of Mexico (INEGI) [17] do not distinguish between "on foot" or "cage" systems. Therefore, these values refer to the average production systems of medium-scale farms for egg production [17]. [e] Calculations by the authors with data from FAO [7] and INEGI [18], see Table 2.

The nutritional values of animal products include the protein content. For milk, the protein content per kilogram is very low in comparison with that of meats and eggs because of the high water content. To correct for the nutritional value, we calculated the LRAP in both per kilogram of animal product and per gram of protein of animal product. To do this, we used a protein conversion factor (Table 5) based on the national consumption of each animal product given by FAO (Food Balance Sheets from FAO [7]). These conversion factors were calculated by dividing the national average per capita consumption of each animal product (kg cap$^{-1}$ year$^{-1}$) by the average protein consumption of the same animal product (g protein cap$^{-1}$ day$^{-1}$), converting it to kilograms per year. These values were used to calculate the LRAP in terms of protein (Table 6).

**Table 5.** Nutritional value in protein content of animal products. Calculations based on FAO [7] data for the 2013 average national consumption of product in kg cap$^{-1}$ year$^{-1}$ and protein consumption (g protein cap$^{-1}$ day$^{-1}$).

| Animal Product | Conversion Factor [g Protein × kg Animal Product $^{-1}$] |
|---|---|
| Beef | 148 |
| Milk | 32 |
| Pork | 108 |
| Chicken meat | 104 |
| Eggs | 101 |

**Table 6.** Animal products consumption per capita for the present average Mexican diet and the USA diet. Source: Food Balance Sheets, values of food supply per capita in 2013 from FAO [7].

| | Present Mexican Diet [kg cap $^{-1}$ year $^{-1}$] | Present USA Diet [kg cap $^{-1}$ year $^{-1}$] |
|---|---|---|
| Beef | 15 | 36 |
| Milk | 112 | 255 |
| Pork | 15 | 27 |
| Chicken meat | 30 | 50 |
| Eggs | 18 | 15 |

*2.6. Estimation of the Land Demand for Food for the Mexican Population*

In Section 3.2, the results of the LRAP were scaled up to assess the total land demanded by the Mexican population considering all the food consumed by the Mexican population in the present and in the future; this includes the land to produce both the animal products and the crop-based diet. The land demanded for the consumption of the animal products can be estimated by multiplying the LRAP (Table 1) by the per capita consumption of each of the five animal products (Table 6). These values were multiplied by the LRAP and the results are shown in Table 7. For milk production, the grazing system was used, and for egg production, the chicken on foot system was used.

**Table 7.** Land use for the food consumption in Mexico. The first row shows the total population of Mexico. The rest of the rows show the values of Land use for food consumption for the total population of the country. See text for details.

| | Present | | 2050 with Present Diet | | 2050 with Affluent Diet | |
|---|---|---|---|---|---|---|
| | **Grassland** | **Cropland** | **Grassland** | **Cropland** | **Grassland** | **Cropland** |
| People [Million people] | 123 | | 164 | | 164 | |
| **Land use** [Million hectares] | | | | | | |
| Beef | 66 | 4 | 89 | 5 | 209 | 12 |
| Milk | 12 | 3 | 16 | 4 | 36 | 8 |
| Pork | - | 2 | - | 3 | - | 5 |
| chicken meat | - | 7 | - | 9 | - | 15 |
| Eggs | - | 1 | - | 2 | - | 1 |
| Crop-based products | - | 13 | - | 18 | - | 18 |
| Total land use | 78 | 30 | 104 | 40 | 245 | 60 |

For the crop-based products, we used the data of Kastner et al. [28], who calculated the land requirements for food for all subcontinents in the world between 1960 and 2007. The land for the crop-based products at the end of the study period ranged from 900 to 1200 m$^2$ cap$^{-1}$ year$^{-1}$, thus, we assumed that 1100 m$^2$ cap$^{-1}$ year$^{-1}$ were needed to produce the crop-based portions of the diet.

The beef, milk, pork, chicken meat, and egg consumption accounts for 84% of the total animal protein consumption in the diet in Mexico, thus, these values are a good estimate of the total land demand for the total food consumption. Values for the individual land demand per person for each of the five animal products and for the crop-based products (Table 7) were added and multiplied by the total population to derive the total land use for food consumption in Mexico. Three scenarios were considered (Table 7): (1) the present situation with 123 million people (2013), (2) the situation in 2050 with population growth to 164 million people and no dietary changes (assuming the average diet of 2013, see Table 6), and (3) the situation in 2050 with population growth to 164 million people and dietary changes to affluent consumption.

To assess future dietary changes towards an affluent consumption, we considered a high estimate scenario for 2050 by assuming that the Mexican population will have an average diet resembling the present consumption in the USA (Table 6). This assumption is based on studies that have shown that dietary patterns are changing to affluent diets including a relatively important increased consumption of animal products [28,29]. Diets are not changing equally throughout the world [29]: for instance, the consumption of animal products per capita increased rapidly between 1960 and 2010 in China and South America; however, in China, the increase was in pork, and in South America, the increase was in chicken. With regional characteristics in mind, we assumed that in Mexico the change to an affluent consumption in the coming years might follow the USA pattern due to the similarity in the type of animal products that the two countries consume (Food Balance Sheets from [7]) and the strong influence and proximity between the two countries.

*2.7. Sensitivity Analysis of the Production Variables*

The values of the production variables used in this paper (Table 4) were obtained from the average of a medium-scale farm in Mexico and from recommendation manuals for Mexican farmers. These values vary widely in the literature owing to differences in production systems and in local biophysical conditions. A sensitivity analysis was performed to (1) analyze the role of the production variables (Table 4) on the LRAP to identify options to reduce it and to (2) validate the values of the production variables used in this paper and to determine whether our assumptions have an effect, in orders of magnitude, on the differences in land requirement among animal products. The high and low estimates of the production variables that are used to perform the sensitivity analysis (Table 8) illustrate the

variations of the values that exist in the literature. The LRAP for the sensitivity analysis was calculated by using the values of Table 4 except for each value of the high and low estimate of Table 8.

**Table 8.** Values of the production variables to perform the sensitivity analysis.

| Higher and Lower Estimates of the Production Variables | Beef | Milk | | Pork | Chicken Meat | Eggs | |
|---|---|---|---|---|---|---|---|
| | | Grazing | In Barn | | | On Foot | Caged |
| More barn area per LU: 50% more [m² LU⁻¹] | 18 | 18 | 18 | 1.59 | 0.135 | 2.085 | 0.705 |
| Less barn area per LU: 50% less [m² LU⁻¹] | 6 | 6 | 6 | 0.53 | 0.045 | 0.695 | 0.235 |
| More efficient feed crops: 50% higher crop yield [ton ha⁻¹] | 10.95 | 10.95 | 10.95 | 6 | 6.75 | 6.75 | 6.75 |
| Less efficient feed crops: 50% lower crop yield [ton ha⁻¹] | 3.65 | 3.65 | 3.65 | 2 | 2.25 | 2.25 | 2.25 |
| Higher carrying capacity of Pastures: 50% more [LU ha⁻¹] | 0.3 | 0.705 | – | – | – | – | – |
| Lower carrying capacity of Pastures: 50% less [LU ha⁻¹] | 0.1 | 0.235 | – | – | – | – | – |
| Higher final live weight: 20% more [kg] | 646.8 | – | – | 105.6 | 2.76 | – | – |
| Lower final live weight: 20% less [kg] | 431.2 | – | – | 70.4 | 1.84 | – | – |
| Higher dressing factor: 20% higher | 0.468 | – | – | 1 | 0.924 | – | – |
| Lower dressing factor: 20% lower | 0.312 | – | – | 0.72 | 0.616 | – | – |
| Higher daily food production [kg-food LU⁻¹ day⁻¹]: 50% higher | | 10.05 | 21 | | | 0.105 | 0.105 |
| Lower daily food production [kg-food LU⁻¹ day⁻¹]: 50% lower | | 3.35 | 7 | | | 0.035 | 0.035 |

## 3. Results

### 3.1. Land Requirements for Animal Products (LRAP)

The share of each type of land in the total LRAP differs markedly (Table 9). Grassland is used only in beef and milk production, and in grazing systems for cattle its contribution to LRAP is one order of magnitude higher than that of cropland, which in turn is two orders of magnitude higher than that of barn area. For pork, chicken, and eggs, cropland has a greater role in the LRAP than has barn area.

**Table 9.** Contributions of three types of land use to the Land Requirement for Animal Production (LRAP) in Mexico, and to the protein content of the animal products.

| Animal Product | Land Use | Per Weight [m² year kg-food⁻¹] | Per Protein [m² year kg-protein⁻¹] |
|---|---|---|---|
| Beef | Grassland | 351.5 | 2374 |
| | Cropland | 20.3 | 137 |
| | Barn | 0.11 | 0.8 |
| Milk (grazing system) | Grassland | 8.6 | 264 |
| | Cropland | 1.9 | 59 |
| | Barn | 0.005 | 0.2 |
| Milk (barn system) | Cropland | 4.8 | 147 |
| | Barn | 0.005 | 0.2 |
| Pork | Cropland | 11.8 | 109 |
| | Barn | 0.007 | 0.1 |
| Chicken meat | Cropland | 18.3 | 176 |
| | Barn | 0.012 | 0.1 |
| Eggs (Chickens on foot) | Cropland | 5.3 | 53 |
| | Barn | 0.054 | 0.5 |
| Eggs (Caged chickens) | Cropland | 5.3 | 53 |
| | Barn | 0.018 | 0.2 |

Beef requires 23–35 times more land to produce one kilogram of meat than do pork, chicken meat, or milk (in grassland systems), and 80 times more land than eggs or milk (in barn systems).

In terms of protein production, the land requirement for beef is seven times higher than for milk in grazing systems, 17 times higher than for milk (in barn systems) and chicken meat, 24 times higher than for pork, and 54 times higher than for egg production. Beef is the animal product with the highest protein content per kilogram of food, followed by eggs, pork and chicken meat, and milk has the lowest protein content (Table 5). The low protein content of the milk is due to its large water content. Thus, in terms of protein, the land requirements per animal product differ (Table 9). For instance, milk in barn systems and chicken meat require a similar amount of land, more than for pork production. However, even though beef has the highest protein content, it also has the greatest land requirement in terms of protein, this being one order of magnitude higher than for the other animal products.

The high requirement of land for beef is driven by a mixture of variables (Table 4) for which cattle are less efficient in terms of production than are chicken and pigs. These inefficiencies outweigh the fact that the final live weight of the cattle is much higher than that of pigs (six times) or chicken (234 times) (Table 4): (a) cattle use 10 times more land for housing (barn area) than pigs and chickens; (b) cattle consume 7.7 times more feed in their lifetime than pigs, and 202 times more than chickens; (c) the lifespan is 2 years for cattle, 6 months for pigs and 2.8 months for chickens; and (d) the dressing factor for cattle is very low (0.39) compared with the dressing factor of chickens (0.77) or pigs (0.9).

The use of grassland has the largest role in the total LRAP for beef production. This is due to both the low LU density of the grassland (only 0.20 LU per hectare) and the long time that bullocks are raised on grassland (1.5 years). Cows for milk production also require grassland, but the amount of grassland needed per protein unit by weight is nine times smaller than the amount of grassland per beef protein unit. Land for milk and beef production is driven by a complex mix of production variables: land for beef depends on the land that the animal uses during its lifetime (and on its final live weight and dressing factor), whereas the grassland area for milk depends on the cows' productivity and the land that the cow uses during that time. Overall, the production of a beef protein unit requires eight times more land than the production of a milk protein unit, indicating that the production of beef protein is much less efficient in terms of land than the production of a milk protein in a grassland system (Table 9).

Comparing the two milk production systems, the grassland system requires 2.3 times more land to produce 1 milk protein unit than does the barn system. This difference is driven by three variables (Table 4): grassland is used in grazing systems and not in barn systems; feed consumption is 2.4 higher in barn systems than in grazing systems; and milk production is twice as high in barn systems. This shows that the role of grasslands in the total LRAP outweighs the other variables and that an increased use of cropland does not necessarily increase the total LRAP, because greater feed consumption increases the cow's daily milk production. However, further studies should consider the differences between grasslands and croplands in terms of quality and production potentials (see Section 4.3).

For egg production, although the barn area per chicken in the "chicken on foot" system is three times higher than in the "caged chicken" system (Table 4), the difference in the total LRAP between these two systems is very low. Hence, the requirement for cropland outweighs the requirements for barn area.

*3.2. Land Demanded by the Present and Future Mexican Population*

The total land demand for the food consumption in Mexico depends on three factors: (1) the land needed per kilogram of food produced (Table 9 for the animal products, and Kastner et al. [28] for the crop-based products); (2) the total population of the country; and (3) the per capita consumption of food (Food Balance Sheets from FAO [7]). Section 2.6 shows the calculation to estimate the land demand for the Mexican population. The Land demand includes the land used for food consumed in Mexico both from national production and from food imports; the land available includes the land used for food production for national food supply and for food exports. When the land demand by the

Mexican population for food is compared with its availability (Table 10), this can indicate the potential for food sufficiency in Mexico and the need for food imports.

**Table 10.** Land demand versus Land available. The land available in 2013 and 2050 is assumed to be the same. Source of data: Land demand, calculations by the authors (see text for details); Land available, from FAO [7], data of "Permanent meadows and pastures" for grassland, and "Arable land and permanent crops" for cropland.

|  | **Land Demand** | **Land Available** |
|---|---|---|
| Present (2013)<br>(123 Million people) | Grassland: 78 Mha<br>Cropland: 30 Mha | |
| 2050 with present diets<br>(164 Million people) | Grassland: 104 Mha<br>Cropland: 40 Mha | Grassland: 81 Mha<br>Cropland: 26 Mha |
| 2050 with affluent diets<br>(164 Million people) | Grassland: 245 Mha<br>Cropland: 60 Mha | |

Mexico is presently using 81 Mha of grassland. According to our calculations, only 78 Mha is used for domestic consumption (Table 10), suggesting that the produce (beef, milk, and eggs) from 3 Mha of grassland is exported; this concurs with the values of export statistics [7]). In contrast, for cropland, 30 Mha is demanded by the present Mexican population, while only 26 Mha is available (Table 10). This indicates that a share of the food is produced in other countries; this concurs with the values of import statistics [7].

It is assumed that future food demand in Mexico will increase because the population will grow, and diets will change to a more affluent consumption. In 2050, the Mexican population will be 164 million people (assuming an average fertility rate, see "Population" from [7]). Land demand for this number of people will be 104 Mha of grassland and 40 Mha of cropland, if diets remain stable. These values are 30% and 50% higher than the present available grassland and cropland, respectively. However, land demand might be higher than this value owing to dietary changes. For instance, by 2050, the Mexican population could have the USA average consumption of animal products (2.3 times the present per capita beef consumption and 2.3 times the per capita milk consumption, which is a high estimate); then, the land demand would increase to 245 Mha of grassland and 60 Mha of cropland. This is triple the present available grassland and twice the present available cropland in Mexico. Hence, the changes in diets will have a higher impact on land demand than will population growth. See Section 2.6 for details of the differences in diets.

*3.3. Role of the Livestock Production Variables on Changing the LRAP*

A sensitivity analysis (Table 11) shows that the orders of magnitude of the land requirements among animal products is not affected by a low or high estimate (Table 8) of the livestock production values. Hence:

(1)　The land requirement to produce 1 kg of beef is at least one order of magnitude higher than for the other animal products.
(2)　The land requirement to produce 1 kg of eggs is always the lowest.
(3)　The land requirements to produce 1 kg of milk, chicken meat or pork are similar, and differ mainly according to the production efficiency for the animals (dressing factor or daily milk production per animal) and the crop yield of the feed.

**Table 11.** Sensitivity analysis of the production variables to calculate the LRAP. Values expressed as area required to produce 1 kg of food [m$^2$ year kg-food$^{-1}$]. In parenthesis: deviation from the LRAP calculated in this paper. Deviation calculated by dividing the difference between LRAP of the assumption and the LRAP calculated in this paper by the LRAP calculated in this paper.

| Higher and Lower Estimates of the Production Variables | Beef | Milk | | Pork | Chicken Meat | Eggs | |
|---|---|---|---|---|---|---|---|
| | | Grazing | In Barn | | | On foot | Caged |
| LRAP values of this paper | 371.91 | 10.500 | 4.759 | 11.787 | 18.301 | 5.40 | 5.36 |
| More barn area per LU: 50% more | 371.97 (+0.0002) | 10.503 (+0.0002) | 4.762 (+0.0005) | 11.790 (+0.0003) | 18.307 (+0.0003) | 5.42 (+0.005) | 5.37 (+0.002) |
| Less barn area per LU: 50% less | 371.86 (−0.0002) | 10.498 (−0.0002) | 4.757 (−0.0005) | 11.783 (−0.0003) | 18.295 (−0.0003) | 5.37 (−0.005) | 5.35 (−0.002) |
| More efficient feed crops: 50% higher crop yield | 365 (−0.018) | 9.8 (−0.06) | 3.2 (−0.33) | 7.9 (−0.33) | 12.2 (−0.33) | 3.6 (−0.33) | 3.6 (−0.33) |
| Less efficient feed crops: 50% lower crop yield | 392 (+0.055) | 12.4 (+0.18) | 9.5 (+0.99) | 23.6 (+0.99) | 36.6 (+0.99) | 10.7 (+0.98) | 10.7 (+0.99) |
| Higher carrying capacity of pastures: 50% more | 254 (−0.31) | 7.6 (−0.27) | – | – | – | – | – |
| Lower carrying capacity of pastures: 50% less | 723 (+0.95) | 19.1 (+0.82) | – | – | – | – | – |
| Higher final live weight: 20% more | 317 (−0.15) | – | – | 9.8 (−0.17) | 15.3 (−0.17) | – | – |
| Lower final live weight: 20% less | 465 (+0.25) | – | – | 14.7 (+0.24) | 22.9 (+0.25) | – | – |
| Higher dressing factor: 20% higher | 312 (−0.16) | – | – | 10.6 (−0.1) | 15.3 (−0.16) | – | – |
| Lower dressing factor: 20% lower | 469 (0.26) | – | – | 14.7 (+0.25) | 23.0 (+0.26) | – | – |
| Higher daily food production per LU: 50% higher | – | 7.0 (−0.33) | 3.2 (−0.33) | – | – | 3.6 (−0.33) | 3.6 (−0.33) |
| Lower daily food production per LU: 50% lower | – | 21.1 (+1.01) | 9.6 (+1.01) | – | – | 10.8 (+1.00) | 10.7 (+1.00) |

The magnitude of the deviation between each estimate and the LRAP based on the assumptions of this paper (Table 4) indicates which production variables have a greater or lesser role in changing the LRAP (Table 11).

For beef production, the carrying capacity of the pastures (in LU per area) has the strongest role in the LRAP. An increase of 50% in LU per area can reduce by one-third the LRAP (−0.31, Table 11). However, lowering the carrying capacity by 50% can almost double the LRAP (+0.95, Table 11). Next in importance are the final live weight of the cattle and the dressing factor. The role of crop yield is less important. For all animal products the number of animals per barn area has a very low impact on the LRAP

For milk production in grazing systems, increasing daily milk production of the cows has the highest potential to reduce the LRAP (−0.33), followed by the carrying capacity of the pastures (−0.27). The crop yields of the feed have a greater role in the LRAP in the milk and eggs production than in the beef production (−0.06 versus −0.018); this is attributable to the higher feed consumption by dairy cows in these production systems than by beef cattle. For milk production in barns, both the daily milk production per cow and the crop yields of the feed have the greatest role in the LRAP.

For pork, chicken meat, and eggs, the crop yields have the greatest role in the LRAP. Hence, identification of the source of feed for the livestock farm is essential in any discussion of the LRAP.

## 4. Discussion

### 4.1. Option to Reduce LRAP

Strategies to reduce the LRAP should focus on the drivers that have the strongest role in the LRAP as identified in Section 3.3.

Hence, the main focus should be on the reduction of grasslands, then on the reduction of cropland for feed production; lower priority should be given to the reduction of barn area, because the demand for grasslands outweighs the demand for cropland for feed, and the latter outweighs the demand

for barn area (Table 9). Nevertheless, a further assessment should consider the quality of land, since grasslands have much lower quality, and therefore, lower production potential per m$^2$, than croplands (see Section 4.3).

In relation to the reduction of grasslands, increasing the carrying capacity of the grasslands (thereby increasing LU density) has the greatest potential to reduce LRAP for beef production and milk with grazing systems (Table 11).

In relation to the reduction of cropland, the values of the crop yields have a strong role in the LRAP mainly for milk, chicken meat, and pork production (Table 11). The feed should consist of crops that have high yields. Further assessment should consider the trade-offs between increased crop yields and the environmental effects of fertilizers, irrigation, herbicides, and other measures [30]. In addition to the direct drivers for the reduction of grasslands and crop land for feed, other production variables should be considered which increase the efficiency, in terms of production, in the use of land. Larger dressing factors, larger final live weight of the animals, and higher productivity (daily milk and egg production per animal) have an important role in the reduction of the LRAP (Table 11).

Although considerations of animal welfare might require an increase in barn area per animal [31], this might not increase significantly the total LRAP since the barn area is several orders of magnitude lower than cropland and grasslands.

Consumption drivers, too, can play an important role in the strategy to reduce land use for food. A change in the protein consumed, from beef to pork or from beef to chicken, would reduce LRAP (Table 9) [3], and there are wide differences in the land requirements between diets with high and low (or no) consumption of animal products [1–3]. The change in type of animal product should focus on choosing livestock animals with high dressing factors, low lifespans, and high feed-food conversion (or lower feed requirements).

Thus, an integral strategy considering production and consumption drivers should include the factors that have the strongest role on the determination of the LRAP.

### 4.2. Our Results Compared with the Literature

A review by Nijdam et al. [12] of 52 studies that calculated the land requirements to produce animal products using the Life Cycle Assessment approach in western and northern Europe, USA, Canada, Australia, and Japan showed that the land requirements for beef production range between 7 and 420 m$^2$ kg$^{-1}$, for milk between 1 and 2 m$^2$ kg$^{-1}$, for pork 8–15 m$^2$ kg$^{-1}$, for chicken 5–8 m$^2$ kg$^{-1}$, and for eggs 4–7 m$^2$ kg$^{-1}$. Our values (Table 6) fit within these ranges, except for milk production, in which our systems require 5–10 m$^2$ kg$^{-1}$. This difference might be because the systems analyzed in Nijdam et al. for milk production were in Western and Northern Europe and in Canada, where land use is more intensive than in the Mexican milk production systems.

The review by Nijdam et al. shows that beef production systems vary widely according to the type of production system (industrial intensive systems, meadow systems, and extensive pastoral systems). The beef system of the present paper fits within the extensive pastoral system, which is mainly driven by the low livestock density of our system: 0.20 LU ha$^{-1}$. The national agency for environment and natural resources of Mexico [32] reports a "theoretical average grassland coefficient" (number of hectares that can support a livestock unit (LU)) in all Mexican states depending on the biophysical and climatic conditions; on this basis, the average number of Livestock Units per hectare ranges from 0.2 in the north of the country (Baja California Sur) to 0.56 in the south of the country (Chiapas). The Mexican states with the largest bovine production are Veracruz, Jalisco, and Chihuahua [17], which have a theoretical animal density of 0.55 LU ha$^{-1}$, 0.12 LU ha$^{-1}$ and 0.05 LU ha$^{-1}$, respectively [32]. The LRAP might differ markedly among these states because of the different LU density that can be reached. Regional differences within the country should be considered in further research to discuss the differences in the total LRAP.

In general, other studies have shown that the land requirement for pork is greater than the land requirement for chicken meat [7,12]. However, in our study, pork required less land than chicken

(11.2 m$^2$ kg$^{-1}$ and 15.9 m$^2$ kg$^{-1}$, respectively). One difference is that the dressing factor for pigs in our study is larger than in these other studies; for instance, Elferink and Nonhebel [8] showed a dressing factor of 0.81 for pigs and 0.75 for chicken. Therefore, our dressing factor, calculated from two data sources [7,17], might be an overestimate. Our results suggest that the average pork systems in Mexico are more intensive than chicken systems. This complexity of dependent variables suggests that local studies are needed to discuss differences among chicken and pork, and among production systems (such as size of farm, productivity, and type of production).

Whereas our study considered a low dressing factor for beef (0.41), other studies show higher dressing factors for beef: 0.59 in the Netherlands [8] and 0.55 for Mexico [32]. Nevertheless, Gasque-Gómez [33] states that the dressing factor for beef production can be as low as 0.35 (or as large as 0.75).

### 4.3. Limitations of Our Approach

#### 4.3.1. Differences among Grasslands and Croplands

In this paper, the area required to produce one kilogram of animal product was compared indistinctly whether it was cropland, grassland or barn area. Nevertheless, strong differences exist between grassland and cropland in relation to management practices, use of agricultural resources, and production potential. Below, we discuss some implications of our approach which did not consider these differences.

(1) Differences in management practices of croplands and grasslands. Usually, croplands have higher use of agrochemicals than grasslands, resulting in different environmental impacts. For instance, in Latin America, the use of fertilizers on grasslands is only 2% of the total use of fertilizers in agriculture [34]. Hence, the differential use of agricultural resources between croplands and grasslands should be considered when comparing the area required to produce one kilogram of animal product.

(2) Potential of grasslands for food production. The grass produced in grasslands is not edible for human consumption, but ruminant animals, including cattle, are efficient converters of grass into human food [35]. O'Mara [35] shows the high potential of grasslands for food security and states that management improvements in grassland systems could increase food production with an efficient energy conversion (considering $CO_2$ emissions). In addition, grasslands confer a wide variety of Ecosystem Services on the local environment and population [35]. Thus, grasslands have more benefits than croplands, as long as management practices are adequate, which should be considered when comparing grasslands and croplands.

(3) Environmental impact of grazing systems. In Mexico, grassland systems are mainly extensive free-grazing systems. In the bovine systems studied in this paper, 80% of the cattle of the medium-scale beef production system are free grazing, and only 20% are subject to "controlled" grazing [18]. The problems with extensive grazing systems are usually related with overgrazing [36]. Cattle on rangelands (free grazing) are usually grazing indiscriminately on native vegetation. Some studies of Mexican pastoral systems have shown deleterious effects on the local vegetation usually attributable to poor management of cattle [37] and other ruminant animals such as goats and sheep [38]. Hence, improved management practices could reduce local damage and increase food production. Further studies should design options for management improvements to increase the efficiency of land use in grazing systems. These options should be specific to the local biophysical characteristics of the region and the local management practices.

#### 4.3.2. Data Source and Scale of Our Results

The production data from the Agricultural Survey used in this paper [17] are farm-scale data aggregated to a national scale; this has yielded a large sample of farms to calculate an average value in order to obtain insights into the national overview of livestock systems. Nevertheless, the limitation of

these values lies in the possibility of deviation from local values due to the heterogeneity of the country in terms of climate and management practices. Further research is needed to calibrate our results with data from case studies of livestock farms mainly in terms of feed consumption per animal, sources of feed, feed crop yields, dressing factors, animal lifespan, and number of animals per unit pasture area.

## 5. Conclusions

The present food consumption in Mexico requires 108 Mha of land. The largest share involves the use of grasslands for beef and milk production. The expected increase in population and changes in the consumption patterns in the near future could increase the demand for land to >300 Mha. The largest share of this increase is due to the changes in consumption patterns (to require more animal products). This area of land is not available within the Mexican borders. Options to be considered include an increase in the productivity of grasslands and prudence in the choice of diet.

**Author Contributions:** Calculations and data analysis by M.-J.I.-R.; conceptualization, design of methodology, discussion of results and writing of the paper by M.-J.I.-R. and S.N.

**Funding:** This study was partly funded by the program "Programa de Apoyo a Proyectos de Investigación e Innovación Tecnológica (PAPIIT)" of UNAM in Mexico with project number: IA300219.

**Acknowledgments:** The authors gratefully thank Ann Grant for doing careful English proofreading of the paper. The data used in this paper were processed and analyzed with the project number LM-530 at the Microdata Laboratory of INEGI at their offices in Mexico City.

**Conflicts of Interest:** The authors declare no conflict of interest.

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
