# Peer review of "Does Mexico Have Enough Land to Fulfill Future Needs for the Consumption of Animal Products?"

_agriculture, doi:10.3390/agriculture9100211_

Round 1
Reviewer 1 Report
Dear Authors,
I would like to thank you for taking my previous comments into consideration. Now, the revised version of the manuscript is much improved and I believe that the paper could be published in Agriculture journal, after a English proofreading by a mother tongue.
Best regards.
Author Response
Thank you for your review. The paper has been proofread by an english native speaker.
Reviewer 2 Report
As complexity of the issue addressed and that further research is needed to refine calculations taking into account the diversity of Mexican production systems, as acknowledged by the authors, I would suggest to be more cautious with the following sentence of the Conclusions section:
"Several options are needed... such as....a change in diets from beef and milk to chicken, pork and eggs."
I'm not convinced this conclusion can be drawn from this manuscript, hence, I suggest to remove or rephrase this part of the sentence, for instance: "...a change in the type of humans diets"
Author Response
Thank you for your comments. The conclusion has been modified as suggested. Also, the paper has been proofreading by an english native speaker.
This manuscript is a resubmission of an earlier submission. The following is a list of the peer review reports and author responses from that submission.
Round 1
Reviewer 1 Report
New comments are provided in blue and are underlined
- It is not likely that no protein concentrate feed sources are used (produced/imported) in Mexico for livestock feeding, especially for pig and poultry production (for instance, soya meals, which is widely used worldwide). Please, check this relevant issue.
You are right. According to the FAO, the main feed crops used in Mexico are “maize and products” (58%), “sorghum and products” (34%), soybeans (4%) and Barley (see: http://www.fao.org/faostat/en/#data/FBS). So, soybeans are a protein-rich feed crop which is relatively important in Mexico. Most of it comes from imports (see table 1). In our data sources, soya in mentioned as part of the feed mix for pigs and chicken, but since its share is low, in the first version of this paper we excluded it in our calculations and we assumed that all feed was composed by cereal grains to reduce complexity in the calculations. For instance, our source of data for pig’s feed (Lesur 2003 “Manual de Porcicultura”) specifies that pig’s feed should be 80% sorghum, 10% of soya, and the rest are minerals and others; and the data source for chicken (Lopez 2005) specifies that chicken feed should consist of 66% maize, 13% soya and 4% of others. In the new version of the paper, we changed this assumption to include the soybeans. We wanted to explore whether including this would impact the LRAP since its crop yield is lower than the cereal grains, and therefore the land per kilogram of feed will be higher (see table 1). The new values of LRAP are in the same order of magnitude that the first version of the paper, but a little larger because of the lower crop yields of the soybeans.
For bovine feed, we searched several sources and we found that the main feed concentrates used in Mexico for cattle are based on cereals, we did not find a recommendation which include soybeans or other legume. Our data source for bovine (Lesur 2005) specifies that the feed is generally a mix of maize, sorghum and barley (which is what we used in the calculations), it also mentioned that some concentrates are richer in protein content based on oats, maize and wheat, which are also cereals. So, for the bovine feed we did not do any changes.
Authors’ response is not convincing.
In http://www.fao.org/faostat/en/#data/FBS FAO data, only soybeans data are provided and soya meal data are lacking. In Mexico, 5,000,000 t of soya meals are used, according to https://www.agrosintesis.com/la-industria-la-soya-mexico/#.XPjnV7gYEfw (“pasta soya”), probably most of them used as feed mix ingredient. Lesur 2003 and Lopez 2005 are not enough references for animal production data source.
- References on animal production data, particularly no. 15 and 21-26, are in Spanish and not available for international readers. Therefore, it is not possible to assess if the data used from these references are scientifically obtained, representative enough, and account for diversity of livestock farming systems in Mexico, particularly ruminant systems. For instance, low dressing factors and stocking rates for cattle are used, as the authors recognize.
Our main data source is the National Agricultural Survey of 2014 of Mexico. We included the sample of this survey of all “medium-scale” farms in the country. By doing this, the values are representative of at a national level. When the survey did not collect production data needed to calculate the LRAP, we used handbooks for livestock systems which were available at the library of the Veterinary Faculty of UNAM (National University of Mexico), in Mexico City. We believe it is a reliable data source that illustrate the general characteristics of Mexican Livestock systems for the variables for which the survey did not provided the data. We added a new paragraph in section “2.2 Data sources” to describe these data sources and for what purpose were they used.
References 16, 19-23 are not enough for animal production data source.
- Line 65. This assertion requires bibliographic references and data on human consumption.
We added the share of protein per capita intake in 2013 for beef, milk, pork, chicken meat and eggs given by the Food Balance Sheets of the FAO, see the last paragraph highlighted in yellow in the introduction.
It is not justified why per capita protein intake, instead of, for instance energy intake, is used here.
- Line 79. “Grasslands” is a poor terminology to name the non-cropland used for ruminants. Please, use Allen et al. 2011 terminology for example. We are confused by this comment. We revised Allen et al. terminology, and we understood that it is better to use the “grassland” term than using “pasture” (as we did in the first version). We have used “grasslands” throughout the text to be consistent. Line 83. “pasture area”, same comment as in line 79. Same response as above.
- Line 90. Please be more specific: each livestock species, breed, physiological state,….etc. We have specified this.
Where?
- Line 130-131. This argument should be in the Discussion section, and should cite appropriate bibliography. This is not an argument from a reference in the literature. This is one of our results based on illustrating in table 4 the multiple variables that drive the LRAP. We have clarified this by referring to table 4.
Where?
- Lines 221-225. Authors should be more cautious (here and in the Conclusions section), please see the ecosystem services comment above, in the case of ruminants grazing systems. Why not discuss here other type of diets, for instance vegetarian or vegan alternatives? Or less meat but more sustainably produced? We have strength in the discussion section the potential of dietary changes to less-meat diets as ways to reduce LRAP. See the yellow highlighted text in section 4.1
Not sufficiently addressed in the manuscript
- Lines 242-249. If these regional differences have not been taken into account, beef production is penalized once again, in this study. The same for dressing factors used (pigs and cattle). This is discussed in section 4.3
Not sufficiently addressed in the manuscript
- Lines 569-579. Bibliographic references are needed through the two paragraphs
- Lines 626-631. Need appropriate literature citations.
- Line 646: what "mounts" mean? "Showing” is grammatically correct here? Please, consider English native proof reading of the whole manuscript.
- Line 647: reference [35] is referred to “small” ruminants, not to cattle. Please, provide appropriate citation.
- In lines 645-646, you mention cows of free grazing in native vegetation, I suggest “rangelands” instead of “grasslands”, for naming this type of use.
- Quality of the produced protein is not taken into account. This is taken into account by calculating the land per protein for each animal product. See the last column of table 8. For details in the calculation see section 2.5 and table 5.
Quality of the protein (from each type of product, beef, milk, pork, eggs) is not taken into account in the manuscript.
Reviewer 2 Report
Dear Authors,
I confirm that your manuscript entitled “How much land is needed to produce our meat, milk and egg? The case of Mexican Livestock systems” is very interesting. Currently, the land requirements of different type of animal products and related production of proteins are a main topic the world over that needs to high level of the attention.
Moreover, I would like to thank the authors for taking my previous comments into consideration. Now, the revised version of the manuscript is much improved i.e. the structure of the manuscript has been changed, the literature review has been extended, a specific section dedicated to limitations has been added. However, some mistakes and misprints persist e.g.
Lines 435. table 2?
Lines 543-544. "This is three times higher than the available grassland and twice the available cropland. Note that changing to an average USA’s diet implies increasing 2.3 times the per capita beef and milk consumption 2.3 times." Please, explain.
Lines 198 (Table 2), 382 (Table 4), 615. Dressing factor 0.41 or 0.39?
Therefore,
my opinion is that this manuscript need a in-depth proofreading, also in English by a mother
tongue.
Moreover, on the basis of results, I suggest to proceed with more thoroughly studies in the future on this topics e.g. with the comparison between meat production and edible insects production.
Best regards.
Reviewer 3 Report
Missing oxford commas from lines 12, 69, 71, 80, 118, 194, 259, 261, 430
Abstract
Line 12: change to “used as an example”
Line 13: missing commas “chicken meat, and eggs.”
Line 19: I think it should be “changes to affluence patterns”
Introduction
Line 38: type and amount should be plural. “...consume different types and amounts of feed”
Line 46: the sentence that stats on this line should start with “No” and not “Not”
Line 50: type should be plural; meat should not be. “…different types of meat and dairy products..”
Line 53: change to “The average per capita consumption of animal products”
Line 58 and 59: delete “the” from “…period, the total meat production…” and “…per year [14], and the exports of meat…”
Line 68: change from “…which account to 84%...” to “…which account for 84%...”
Materials and Methods
Line 80: change from “This amount of land is, for meat, the land….” To “For meat, this amount of land is the land…”
Line 81-82: change to “that animals use during the production of a certain amount of dairy”
Line 90: change to “so they directly use the land…”
Line 93: change to “a young calf counts as…”
Line 95: chicken should be chickens
Line 102 and 107: I think it should be “differs” instead of “divert”
Line 103 and 105: change to “amount and type” to “amounts and types”
Line 109-111: change to “Available feed on the market can very monthly in price and nutritional value [7] and in the mix of imported crops and domestically produced crops [14].”
Lines 119-121: I think there are two separate statements in this sentence that should be separated. 1) Animals with longer life spans use land, barn or grassland areas, longer. 2) cattle need feed to maintain metabolic functions and to put on weight so the conversion of feed to meat obtained is inefficient.
Line 160: “eggs” should be “egg”
Line 177: change “crop yield of the crop” to “crop yield of each crop”
Line 179-180: change to “...from the USA since it is the main source of feed imports to Mexico [25].”
Line 181: change “from national production” to “produced domestically”
Lines 181-182: change to “...which we assumed to be 47% for sorghum forage green”. What about the dry matter percentages of the other forage crops? These do not appear to be included and it is my understanding that dry matter percentage of the feed is critical for cattle operations, regardless of the crop that feed comes from.
Line 191: chance “none edible materials” to “non-edible materials”
Line 194: change laughter to slaughtered
Line 195: change dressing factor to dressing factors
Line 239: change “...generally consist, first on a grazing period, and then, the animals are sent for some months for fattening in a barn which they only stay in-doors” to “…generally consist of a grazing period followed by a period of some months for fattening during which the animals stay inside a barn”
Line 243: change slaughter to slaughtered
From here on I will only be commenting on substantive issues and not grammar. Please have a native English speaker edit your paper for grammar.
Lines 239-245: this section seems repetitive
Lines 252-254: from this, the following section, and equations 1 and 2 it sounds as though you are using the same calculations regardless of the type of operation the cows came from. Either you need to clarify how you differentiated between farms that only graze and farms that only do in-door barn raising, or you need to explain why you treated all of the cows the same. I think there also needs to be some clarification about how you derived the average area used for each production system. Did you calculate based on individual farms and then average, or create and average farm then calculate. I think calculating for each farm, then averaging, although time consuming, has merit because it could be used to provide a recommendation for which types of practices would better enable the food demand to be met.
Table 4: there do not appear to be any numbers in bold, but some are underlined. The table heading needs to reflect what is actually used in the table.
Results
Lines 455-466 and Table 8: Was any sort of statistical analysis performed on this information? If so it needs to be included. Statements about which system use more land would be strengthened with statistical analysis. This goes for statements made in lines 472-504 as well.
Discussion:
You want to make recommendations on what types of practices should be reduced and what increased, but you do not appear to have done any statistical analysis on your results, which weakens your suggestions. Your analysis also appears to impacts other than land use for animal production. What are the environmental implications of increasing the density of livestock on land? Can you do this without still needing enough land to rotate and avoid degradation of the grassland? What about the environmental implications of bard raised vs. grassland raised cattle? You also admit that grassland used for pasture is often less productive than crop land, so what alternative to grassland meat/dairy production could it be used for? Some of this is mentioned in the limitations section, but I think some reorganization might be in order.
You state that the next step would be to calibrate results with case study data to confirm the factors you included in your equation. I would suggest that this is necessary at this stage. If you have farm level data a small number of case study farms could be looked at, at least to validate the averages you have calculated. It also suggests to me that you aggregated farm data to an average and then calculated land use. I think calculating land use for case study farms then calculating and average land use and extrapolating to the whole of Mexico would be a stronger analysis.